

# Climate uncertainty in flood protection planning

Beatrice Dittes[1], Olga Špačková [1], Lukas Schoppa[1] and Daniel Straub[1]

[1] Engineering Risk Analysis Group, Technische Universität München, Arcisstr. 21, 80333 München, Germany

*Correspondence to*: Beatrice Dittes (beatrice.dittes@tum.de)

## Abstract

Technical flood protection is a necessary part of integrated strategies to protect riverine settlements from extreme floods. Many technical flood protection measures, such as dikes and protection walls, are costly to adapt after their initial construction. This poses a challenge to decision makers as there is large uncertainty in how the required protection level will change during the
measure life time, which is typically many decades long. Flood protection requirements should account for multiple future uncertain factors: socio-economic, e.g. whether the population and with it the damage potential grows or falls; technological, e.g. possible advancements in flood protection; and climatic, e.g. whether extreme discharge will become more frequent or not. We focus here on the planning implications of the uncertainty in extreme discharge. We account for the sequential nature of the decision process, in which the adequacy of the protection is regularly revised in the future based on the discharges that have been observed by that point and that reduce uncertainty. For planning purposes, we categorize uncertainties as either
'visible', if they can be quantified from available catchment data, or 'hidden', if they cannot be quantified from catchment data and must be estimated, e.g. from literature. It is vital to consider the hidden uncertainty, since in practical applications only a limited amount of information (e.g. through projections, historic record) is available. We use a Bayesian approach to quantify the visible uncertainties and combine them with an estimate of the hidden uncertainties to learn a joint probability distribution of the parameters of extreme discharge. The methodology is integrated into an optimization framework and applied to a pre-
alpine case study to give a quantitative, cost-optimal recommendation on the required amount of flood protection.

## 1 Introduction

The frequency of large fluvial flood events is expected to increase in Europe due to climate change (Alfieri et al., 2015).
Planning authorities are therefore increasingly looking to discharge projections to assess future protection needs, rather than considering past observations alone. However, projections differ widely in terms of the level and trend of extreme discharge that they forecast. Thus, there is an increasing consensus that future discharge extremes should be modelled probabilistically for flood protection planning (Aghakouchak et al., 2013). This raises two main questions: 1) how does one quantify a relevant uncertainty spectrum and 2) how is this then further used to identify a protection strategy?



Recent studies have aimed at quantifying individual uncertainties in (extreme) discharge (Bosshard et al., 2013; Hawkins and Sutton, 2011; Sunyer, 2014). (Sunyer, 2014) has pointed out the usefulness of finding a framework to combine uncertainties for flood protection planning. The derivation of a probabilistic model of extreme discharge forms the first part of this paper. We quantitatively incorporate climate uncertainty from multiple information sources as well as an estimate of the 'hidden

uncertainty' into learning the probability distribution of parameters of extreme discharge. The term 'hidden uncertainty' refers to uncertainty components that cannot be quantified from the given projections and data. For example, if the same hydrological model has been used for all projections, then the hydrological model uncertainty is 'hidden', since one effectively has only a single sample of hydrological model output. It is vital to consider the hidden uncertainty since in practical applications, only a limited amount of information and models is available.

Once established, the question is then how to deal with the uncertainty in flood risk estimates when conducting flood protection planning. Multiple approaches have been proposed (Hallegatte, 2009; Kwakkel et al., 2010), including the addition of a planning margin to the initial design. The planning margin is the protection capacity implemented in excess of the capacity that would be selected without taking into account the uncertainties. Such reserves are used in practice; for example, in Bavaria, a planning margin of 15 % is applied to the design of new protection measures to account for climate change (Pohl, 2013;

Wiedemann and Slowacek, 2013). Planning margins are typically implemented based on rule-of-thumb estimates rather than a rigorous quantitative analysis (KLIWA Klimaveränderung und Wasserwirtschaft, 2005, 2006; Kok et al., 2008).

We have previously proposed a fully quantitative Bayesian decision making framework for flood protection which takes into account the uncertainty in the parameters of extreme discharge and probabilistically models realizations of future extreme discharges for sequential protection planning (Dittes et al., 2017). The framework recommends a cost-optimal capacity of

flood protection measures given a fixed protection criterion (such as the 100-year flood), taking into account possible future measure adjustments. To protect for the 100-year flood is common European practice (Central European Flood Risk Assessment and Management in CENTROPE, 2013) and is also the requirement in the case study. Here, we show how to incorporate into the framework the visible uncertainty from an ensemble of climate projections as well as hidden uncertainties that can not be quantified from the ensemble itself but may be estimated from literature. We provide reasoned estimates of the

relevant uncertainties for a pre-alpine catchment, followed by an application of the framework, sensitivity and robustness analysis.

Since there is often a discrepancy between the level of observed past discharge at a specific gauge and the corresponding regional climate projections, we take the commonly used approach (Fatichi et al., 2013; Pöhler et al., 2012) of computing relative rather than absolute values from the climate projections. Here, this means that we find a planning margin $\gamma$ based on

the projection ensemble and uncertainty estimates from literature and then apply it to the estimated absolute protection (100-year flood) from historic records.



It is stressed that this paper focusses on the engineering aspect of planning flood protection under climate change. We aim to demonstrate how different sources of uncertainty can be combined probabilistically to make decisions, taking into account future developments. This is to aid decision making under climate uncertainty, when there are limited data and models available. Some authors advocate not using a probabilistic approach when the uncertainty is very large. This is because of the potential of surprises under large uncertainty (Hall and Solomatine, 2008; Merz et al., 2015; Paté-Cornell, 2011). Instead, they recommend an approach focussed on robustness: the ability of the protection system to work well under a wide range of scenarios. We consider our approach to be complementary: rather than representing a definite recommendation for the study site, it gives an indication of the recommended protection capacity. Expert judgement remains valuable to identify robust protection systems realizing the recommended protection, e.g. by implementing a protection system that consists of several different, possibly spatially distributed, measures. That leads to more robust protection in which floods in excess of the design flood do not quickly lead to very high damages or even failure (Blöschl et al., 2013b; Custer and Nishijima, 2013).

The paper is structured as follows: In Sect. 2, we introduce the pre-alpine case study catchment together with the available data and relevant uncertainties, concluding in an estimate of the hidden uncertainties. In Sect. 3, we show how to combine the different sources of uncertainty to use in the decision framework of  (Dittes et al., 2017). The resulting recommendations are presented and discussed in Sect. 4, together with sensitivity analysis. Finally, a discussion is given in Sect. 5 and conclusions in Sect. 6.

## 2 Uncertainty in extreme discharge in a pre-alpine case study catchment

In this section, we introduce individual components of uncertainty in extreme discharge. This is done on the example of a pre-alpine catchment with a short historic record and a limited set of available climate projections, which do not exhaustively cover the spectrum of climate uncertainties. The resulting problem of planning under uncertainty is typical in practice. We introduce the case study catchment in Sect. 2.1, followed by the available discharge projections in Sect. 2.2. We then move on to describe climatic uncertainties in Sect. 2.3-2.4 and give an estimate of their magnitude for our analysis in Sect. 2.5. We end by introducing the mathematical modelling of uncertainties and the respective uncertainty of model parametrization in Sect. 2.6.

### 2.1 The Mangfall catchment in Rosenheim

Our case study site is the river Mangfall at gauge Rosenheim, shortly before it flows into the Inn river. Rosenheim is a city in Bavaria that has suffered severe flood losses from Mangfall flooding in the past (Wasserwirtschaftsamt Rosenheim, 2014). With an area of 1102 km², the Mangfall is a medium-sized catchment exhibiting a highly heterogeneous topography. Elevations within the catchment range from 443 to 1988 m a.s.l. with a mean value of approximately 1000 m a.s.l., indicating the pre-alpine nature of the river basin. Southern sub-catchments in the Mangfall-mountains are steep and rocky, resulting in a rapid runoff response. On the contrary, northern regions in the Alpine foothills show a more moderate discharge behaviour due to





gentle slopes. Thus, the discharge pattern of the Mangfall combines both characteristics of mountainous and lowland areas (Kunstmann and Stadler 2005; RMD Consult 2016; Magdali 2015).

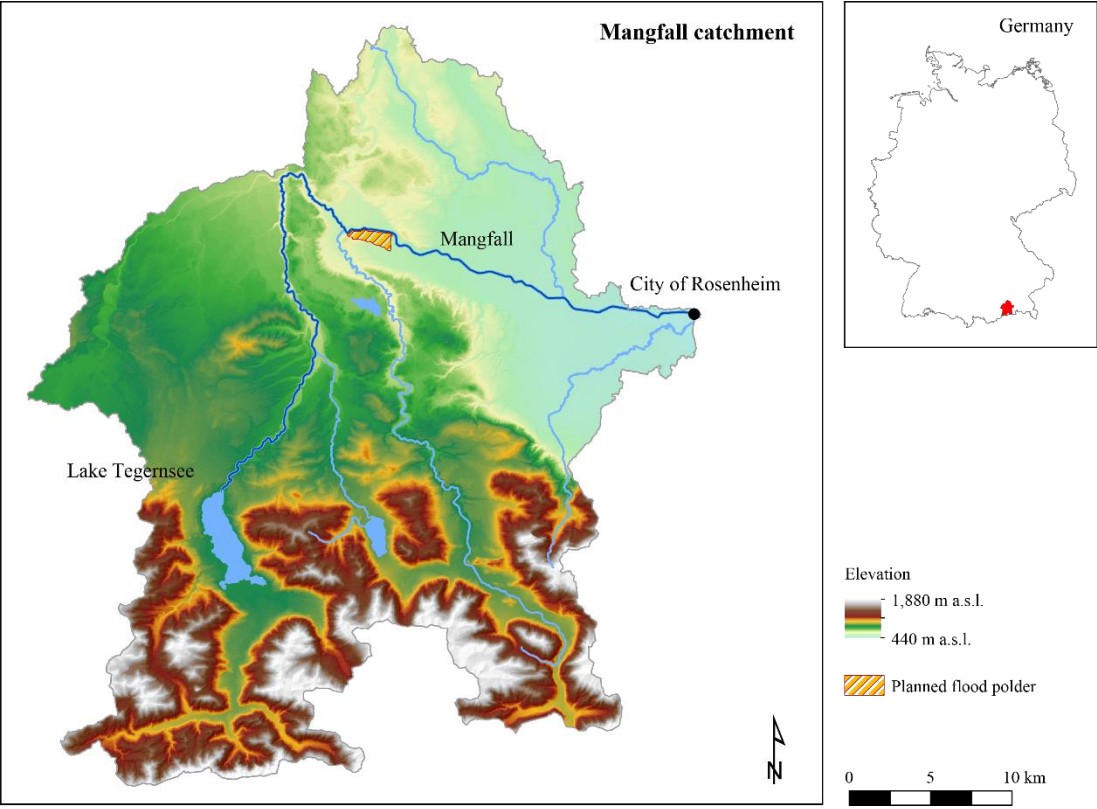

**Figure 1.** Digital elevation model of the pre-alpine Mangfall catchment with its river network. The catchment is
5      characterized by its highly heterogeneous topography leading to different discharge behaviour between the northern and
southern regions of the river basin (Geobasisdaten © Bayerische Vermessungsverwaltung).

Precipitation in the catchment is strongly affected by the adjacent Alpine arch leading to annual mean amounts of 1800 mm in mountainous and 1000 mm in low-altitude zones. The watershed receives most precipitation in July, often in form of
10    convective, high-intensity precipitation (Magdali 2015; Deutscher Wetterdienst 2017). Therefore, this study focusses on the uncertainty analysis for summer discharge, since it poses the greatest threat to the city of Rosenheim. Planning authorities give the 100-year design discharge at the Rosenheim gauge as 480 $m^3s^{-1}$ (RMD Consult 2016). Figure 1 shows the topography of the Mangfall catchment alongside its river network. The available historic record at the Mangfall gauge in Rosenheim is reproduced in Supplement A.



## 2.2 Available ensemble of discharge projections

Table 1 lists the projections available at the case study gauge. Several projections have identical modelling chains and differ only in the model run, six of the ten Regional Climate Models (RCMs) are nested in the same Global Climate Model (GCM),

5    ECHAM5, and all GCM-RCMs are based on the same SRES scenario (A1B), coupled to the same hydrological model (WaSiM) and same downscaling technique (quantile mapping). The ensemble is limited in that it does not cover a wide range of modelling uncertainties, and it is imperfect in that the projections of the ensemble are not independent. The projections are reproduced in Supplement B.

10   **Table 1.** RCMs used in this study, driving GCMs, source of the RCMs, downscaling and hydrological model.

| Name | GCM | RCM | Source | Downscaling | Hydrological model |
|---|---|---|---|---|---|
| CLM1 | ECHAM5 R1 | CLM Consortial | Consortium | Quantile mapping (German federal institute of hydrology BfG), SCALMET (Willems and Stricker, 2011) | |
| CLM2 | ECHAM5 R2 | CLM Consortial | Consortium | | |
| CCLM | HadCM3Q0 | CCLM | ETH | | |
| REMO1 | ECHAM5 R1 | REMO | MPI | | WaSiM v8.06.02, Inn, daily, 1km² |
| REMO2 | ECHAM5 R2 | REMO | MPI | | |
| REMO3 | ECHAM5 R3 | REMO | MPI | Quantile mapping (Bavarian environmental agency LfU), SCALMET (Schmid et al., 2014) | |
| RACMO | ECHAM5 R3 | RACMO2 | KNMI | | |
| HadRM | HadCM3Q3 | HadRM3Q3 | Hadley Centre | | |
| HadGM | HadCM3Q3 | RCA3 | SMHI | | |
| BCM | BCM | RCA3 | SMHI | | |

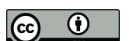



## 2.3 Internal variability

The term 'internal variability' describes the aleatory (alternatively: irreducible) uncertainty component in extreme discharge: even with perfect knowledge, it cannot be predicted with certainty what amount of discharge will be recorded on a given day (Kiureghian and Ditlevsen, 2009). This is because discharge realizations occur spontaneously, due to interactions of components within the climate system (IPCC, 2013). Based on the available information, it can be assumed that the absolute amount of internal variability does not change in time. In projections of future discharge however, the relative importance of internal variability decreases with time as climatic uncertainties increase with increasing projection horizon. In a small pre-alpine catchment, such as considered in our case studies, the internal variability is large and dominates the uncertainty spectrum, potentially masking existing trend signals in heavy precipitation (and thus extreme discharge) for the entire projection horizon up to the year 2100 (Maraun 2013). Alternative terms for the internal variability are 'inherent randomness' or 'noise'.

## 2.4 Uncertainties in the climate modelling chain

Discharge projections are the result of a complex multi-step climate modelling process. In literature, this is often termed the climate modelling 'chain', which, as new uncertainties are introduced at each modelling step, leads to the 'uncertainty cascade' (Mitchell and Hulme 1999; Foley 2010). It is worth pointing out that the uncertainty cascade does not necessarily lead to an increase in uncertainty at each step, as the modelling steps depend on each other in a non-linear fashion. Just as uncertainties can add up, it is conceivable that they may not be relevant for future steps in the modelling chain (Refsgaard et al., 2013). The uncertainty from the interaction of consecutive steps in the modelling chain is called 'interaction uncertainty' (Bosshard et al. 2013).

In the following sections, we briefly introduce the individual modelling steps required to obtain projections of (extreme) discharge. We start by a very brief introduction to climate forcing, then move on to summarise the uncertainty from the global and local climate models (GCM and RCM) under 'Model response'. Statistical downscaling is covered with a focus on quantile mapping, which is the technique applied in the case study. Up to and including statistical downscaling, the climate modelling chain produces not discharge but various other climatic variables that are translated to discharge in a specific catchment through a hydrological model. The uncertainties in the climate modelling chain are in principle epistemic, yet it is debatable if they can and will be reduced in the foreseeable future (Hawkins and Sutton, 2009).

### Forcing

The forcing of the climate through greenhouse gas emissions (GHGs) is the first element in the climate modelling chain. The future socioeconomic, political and technological development determines the amount of GHGs emitted. Different development scenarios on which climate modellers could base their work were described in the IPCC's Special Report on Emissions Scenarios (SRES) (Nakićenović and Swart 2000). Recently, the SRES scenarios were substituted by representative concentration pathways (RCPs), which directly refer to the amount of GHGs emitted rather than complex scenarios (Moss et



al., 2010). For our case study, only projections based on SRES scenario A1B, a widely used scenario with moderate socio-economic and technological changes, are available. Thus, we have to take into account the uncertainty of what the projection results might have been under other forcing scenarios. However, in Europe, forcing uncertainty only becomes relevant in the far future and is of particularly low significance for local extreme precipitation (Hawkins and Sutton, 2011; Maraun, 2013;

Tebaldi et al., 2015).

**Model response**

For climate change impact studies, it is typical to use ensembles of not one but multiple GCM-RCM combinations (Huang et al., 2014; Muerth et al., 2012; Rajczak et al., 2013). The differences in GCM-RCM output when driven by the same emission forcing are termed 'model response uncertainty' or 'model spread' (IPCC, 2013). Multi-model ensembles (MMEs) such as the

one available for the case study reproduce part of this spread. That they do not reproduce it completely is because they consist of a finite number of possibly biased and dependent models that typically have to be chosen based on availability rather than on statistical considerations (Knutti et al., 2013; Tebaldi and Knutti, 2007). To mitigate this problem, some researchers assign weights to individual models, but there is an ongoing debate about this: some researchers are making a general case for the benefits of weighting (Ylhäisi et al., 2015) or its drawbacks (Aghakouchak et al., 2013), some are detailing when it may make

sense on the basis of model performance (Refsgaard et al., 2014; Rodwell and Palmer, 2007) or genealogy (Masson and Knutti, 2011), but all approaches are disputed. The relative importance of model response increases with projection lead time and is particularly significant for extreme summer precipitation (Bosshard et al., 2013). Since flooding in the case study catchment is dominated by extreme summer precipitation, we expect model response to form the second most important uncertainty contribution (after internal variability).

**Statistical downscaling**

The available projections underwent statistical downscaling using quantile mapping, which is often recommended for extreme events (Bosshard et al., 2011; Dobler et al., 2012; Hall et al., 2014; Themeßl et al., 2010). Statistical downscaling is frequently used to align GCM-RCM outputs with historic records, but its use is still controversial (Chen et al., 2015; Ehret et al., 2012; Huang et al., 2014; Maurer and Pierce, 2014). This is particularly because of two key assumptions, which may not always hold

true: that the observational data represents the true state of the climate system and that the bias is stationary (Hall et al., 2014; Korck et al., 2012; Sunyer et al., 2014). The uncertainty contribution of the downscaling is likely to be large (Hundecha et al., 2016; Sunyer et al., 2015a). It would be beneficial to use not one but several downscaling techniques, similarly to how one uses an ensemble of GCM-RCMs (Arnbjerg-Nielsen et al., 2013; Sunyer et al., 2015b), as well as several calibration datasets (Sunyer et al., 2013a).

**Hydrological model**

Hydrological models use RCM outputs such as precipitation, temperature, wind speed and soil moisture to model discharge for a specific catchment. Catchment parameters (such as surface roughness) are typically found in an elaborate calibration



procedure (Labarthe et al., 2014; Li et al., 2012). The parameters are typically assumed to be stationary, but they might in fact be non-stationary (Merz et al., 2011). Furthermore, the calibration might mask model errors by tuning the catchment parameters to balance them. Thus, the parameter estimates strongly depend on the calibration period (Brigode et al., 2013). Several approaches exist to quantify the uncertainty stemming from the hydrological model (Götzinger and Bárdossy, 2008; Velázquez

et al., 2013). Overall however, the error from the hydrological model is small, in particular for high flow indicators (Velázquez et al., 2013). It is likely smaller than or comparable to forcing uncertainty (Wilby, 2005).

## 2.5 Estimate of climatic uncertainty shares in extreme discharge for case study

In this section, we estimate the relative contribution of climatic uncertainties, using internal variability as a reference. To summarise the previous two sections, the following qualitative statements can be made about the contribution of relevant

sources of uncertainty in the considered mid-size pre-alpine catchments with floods driven by summer precipitation:

- internal variability is dominant throughout most of the coming century
- model response is the second largest source of uncertainty, growing with lead time
- the impact of downscaling is also considerable, again particularly later on the projection horizon
- the role of forcing uncertainty and hydrological model is minor; the former becomes relevant only very late on the

15        projection horizon
- uncertainty from interaction of the individual components may be of some significance

A methodology to quantify the size of the internal variability, model response and forcing uncertainty in mean precipitation and corresponding results for different regions and seasons has been presented in (Hawkins and Sutton, 2009, 2011). We base our estimate of these components on equivalent results for summer precipitation in Europe obtained from (Ed Hawkins, email

communication, 17.02.2017). We consider precipitation results to be transferable to discharge in the given catchment since extreme summer precipitation has in the past been the dominant trigger of high discharge in the Mangfall. A comparison of uncertainty shares for mean vs. extreme discharge is available in (Bosshard et al., 2013) and is used to adapt the results. Quantitative estimates of the shares of model response, downscaling, hydrological model and interactions for a different pre-alpine catchment are also provided in (Bosshard et al., 2013). We combine the quantitative results with the catchment-specific

qualitative knowledge to produce the estimate. The uncertainty spectrum is shifted towards the later projection horizon to account for the longer dominance of internal variability in a pre-alpine catchment with small scale, extreme summer precipitation as the flood triggering process. This results in a near-term contribution of the internal variability of at least 80 % of total uncertainty, as expected. The shift also reduces the uncertainty share attributed to model response and emission forcing, which, following (Ed Hawkins, email communication, 17.02.2017), explained over 90 % of total uncertainty by the end of the

century. The shares are adjusted such as to better represent the particular modelling and topography: the share of model response is set to peak at around 40 %. For downscaling, shares of up to 25 % are expected. Uncertainties stemming from interactions are anticipated to lie in the order of 10 %. Contributions attributed to hydrological modelling are set to remain



below 5 % over the whole time horizon. The results of the estimation are shown in Figure 2. Forcing, downscaling, hydrological model and interaction components are 'hidden uncertainties' in the case study. As will be shown in Sect. 3, the sum of hidden uncertainties rather than individual components is used in the Bayesian learning. Thus, it does not matter if the share of any one of these uncertainties has been slightly over- or underestimated. The question of sensitivity will be discussed further in

5    Sect. 4. The estimated variance shares of the 'hidden' uncertainty components and internal variability with respect to total uncertainty for Rosenheim are given in Supplement C.

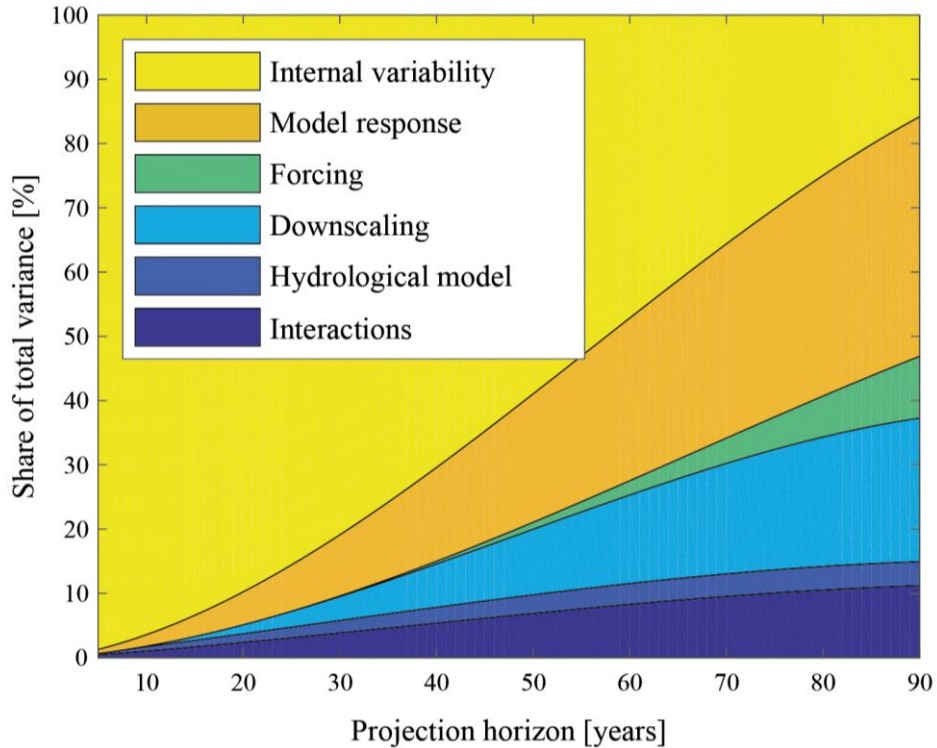

Figure 2. Share of different uncertainty components (variance) for extreme discharge in Rosenheim. Uncertainties that are 'visible' in our case study are shaded yellow/orange, 'hidden' ones blue/green.

## 2.6 Parameter uncertainty

Statistical modelling of extreme discharge $Q$ is commonly based on by fitting a suitable extreme value distribution to the available data, e.g. a Gumbel or a Generalized Extreme Value (GEV) distribution. These are described by their probability density function (PDF), $f_{Q|\theta}(q|\theta)$, in which $\theta$ is the set of parameters of the distribution function that are learned from the

15    data. Learning $\theta$ from finite data will result in a probability distribution over $\theta$, which describes parameter uncertainty (Kennedy and O'Hagan, 2001).


For example, the discharge $q^{(T)}$ of a design flood associated with a return period $T$ is defined as a function of $\boldsymbol{\theta}$ as

$$1 - F_{Q(t)|\boldsymbol{\theta}}\big(q^{(T)}|\boldsymbol{\theta}\big) = \tfrac{1}{T} \quad \leftrightarrow \quad q^{(T)} := F_{Q(t)|\boldsymbol{\theta}}^{-1}\left(1 - \tfrac{1}{T}\Big|\boldsymbol{\theta}\right), \tag{1}$$

where $F_{Q(t)|\boldsymbol{\theta}}$ is the cumulative distribution function (CDF) and $F_{Q(t)|\boldsymbol{\theta}}^{-1}$ is the inverse CDF of the annual maximum discharge $Q(t)$. In a Bayesian framework, the posterior joint PDF of the parameters $\boldsymbol{\theta}$ can be learned from $N$ years of annual maximum discharges $\boldsymbol{q} = [q_1, \dots, q_N]$ (from historic record or projections) as follows:

$$f_{\boldsymbol{\theta}|Q(t)}(\boldsymbol{\theta}|\boldsymbol{q}) \propto L(\boldsymbol{\theta}|\boldsymbol{q}) \times f_{\boldsymbol{\theta}}(\boldsymbol{\theta}), \tag{2}$$

where $f_{\boldsymbol{\theta}}(\boldsymbol{\theta})$ is the prior distribution of the parameters and $L(\boldsymbol{\theta}|\boldsymbol{q})$ is the likelihood describing the discharge data $\boldsymbol{q}$. The discharge maxima can be assumed to be independent between individual years (Coles, 2004). Neglecting measurement error, the likelihood function in Eq. (2) can hence be formulated as

$$L(\boldsymbol{\theta}|\boldsymbol{q}) = \prod_{t=1}^{N} f_{Q(t)|\boldsymbol{\theta}}(q_t|\boldsymbol{\theta}), \tag{3}$$

With increasing number of records of annual maximum discharges $q_t$, the uncertainty in the parameters $\boldsymbol{\theta}$ is reduced.

The Bayesian framework requires the selection of a prior distribution $f_{\boldsymbol{\theta}}(\boldsymbol{\theta})$ in Eq. (2). For the application to flood protection
planning, one may wish to select a prior that is only weakly informative in $q^{(T)}$. We propose to use the following distribution for this purpose (dropping the time dependence $t$ for readability):

$$f_{\boldsymbol{\theta}}(\boldsymbol{\theta}) \propto \frac{1}{f_{Q^{(T)}}(q^{(T)})} = \frac{1}{f_{Q^{(T)}}\left(F_{Q^{(T)}|\boldsymbol{\theta}}^{-1}\left(1 - \tfrac{1}{T}\Big|\boldsymbol{\theta}\right)\right)}, \tag{4}$$

where $f_{Q^{(T)}}\big(q^{(T)}\big)$ is the PDF of $q^{(T)}$ based on a prior that is uniform in $\boldsymbol{\theta}$ and Eq. (1) has been applied in the equality.

## 3 Combining uncertainties for flood protection planning

In this section, we propose an approach for combining different uncertainty components when using projections to learn the parameters $\boldsymbol{\theta}$ of the time-variant distribution $f_{Q(t)|\boldsymbol{\theta}}(q|\boldsymbol{\theta})$ of annual maximum discharge $Q(t)$ in year $t$ (viz. Sect. 2.6). This distribution is first learned for each projection of extreme discharges individually. For projection, we increase the distribution spread in a time-dependent manner using the estimate of hidden uncertainty from Sect. 2.5. Since the uncertainty increases with time, projections late on the horizon are naturally assigned less information value. We then combine the distributions
from different projections.





In Sect. 3.1, we categorize uncertainties in such a way that it is conducive for our application. We then combine these uncertainties within a Bayesian approach. In Section 3.2, we show how the likelihood $L(\boldsymbol{\theta}|\boldsymbol{q})$ for the joint parameter PDF is learned for any individual projection, taking into account uncertainty estimates from literature. In Sect. 3.3, we show how to combine the likelihoods of the projection ensemble. Finally, we give a summary of planning and decision making under

uncertainty in Sect. 3.4.

### 3.1 Uncertainty categorization

Depending on the application different categorizations of uncertainty have been proposed in literature. In Sect. 2.3 - 2.6 for example, we have presented the uncertainties in extreme discharge by source. Another common way to categorize uncertainties is the distinction between aleatory (irreducible) and epistemic (reducible) uncertainties (Kiureghian and Ditlevsen, 2009;

Refsgaard et al., 2013). This categorization is useful in that it underlines in which areas future research could lead to uncertainty reduction. Other authors focus their categorization e.g. on the different effects of uncertainties (Merz et al., 2015). In the context of estimating flood extremes under climate change with limited information, we distinguish between:

- ▪ 'Visible uncertainty', which is known and can be quantified. For an ensemble of discharge projections, this would e.g. be the internal variability, the model response uncertainty and parameter uncertainty. Parameter uncertainty is
also visible in that it is straightforward to quantify, but it is not a climatic uncertainty.
- ▪ 'Hidden uncertainty', which is the remaining uncertainty and can, at best, be estimated. E.g., in the projection ensemble of the case study, forcing uncertainty is hidden since all projections are based on the same emission scenario. In real planning situations, hidden uncertainty is typically significant because of limited and imperfect projections and data, it can therefore not be neglected.

In the following sections, a methodology will be presented to learn the distribution of parameters of annual maximum discharge using these uncertainties.

### 3.2 Accounting for uncertainty and bias in projections

When using discharge projections, it is important to account for uncertainty and bias within them. As discussed in Sect. 2, climatic uncertainties increase with the projection horizon and thus the information value of a projection made late on the

horizon is smaller than that of an earlier one. For example, a projection for the year 2100 is associated with higher uncertainty than one that is made for the coming year and should have less weight when learning the parameters $\boldsymbol{\theta}$ of the distribution of annual maximum discharge from climate projections. In the following, we develop a methodology that accounts for this.

We introduce the standard deviation $\sigma_{i,t}^{(u)}$, in which the superscript (u) describes which type of uncertainty is considered (internal or hidden), the subscript i denotes the projection and the subscript t the time dependence. The internal variability in

a projection, $\left[\sigma_i^{(internal)}\right]^2$, can be quantified following (Hawkins and Sutton, 2009). Note that the subscript t is excluded here




since internal variability is assumed to be independent of time. Relative variance shares of the individual uncertainties, including 'hidden' ones can be estimated using literature (Bosshard et al., 2013; Hawkins and Sutton, 2011) and expert judgement, as was done in Sect. 2.5. The share of an individual uncertainty component in the total variance is here labelled $\eta_t^{(u)}$, with the indexing as for $\sigma$. The uncertainty shares are assumed to be general for a given location, independent of the

projection. Thus, the absolute value of the hidden uncertainty can be found from the absolute internal variability and the uncertainty variance shares of Sect. 2.5 (reproduced numerically in Supplement C) as

$$\sigma_{i,t}^{(hidden)} = \sigma_i^{(internal)} \times \sqrt{\frac{\eta_t^{(hidden)}}{\eta_t^{(internal)}}}. \tag{5}$$

For learning the joint PDF of the parameters $\boldsymbol{\theta}$ of the annual maximum discharge distribution, we treat the $i = 1, \ldots, M$ discharge projections $\boldsymbol{p}_i = [p_{i,t=1}, \ldots, p_{i,t=N}]$ as samples of the true future discharge $\tau_t$ with a bias $\Delta_{i,t}$: $\tau_t = p_{i,t} - \Delta_{i,t}$. We express the likelihood $L_{i,t}(\boldsymbol{\theta}|p_{i,t}, \Delta_{i,t})$ describing the annual maximum discharge of projection i in year t as

$$L_{i,t}(\boldsymbol{\theta}|p_{i,t}, \Delta_{i,t}) = f_{Q(t)|\Theta}(p_{i,t} - \Delta_{i,t}|\boldsymbol{\theta}), \tag{6}$$

where $f_{Q(t)|\Theta}$ is the PDF of the extreme value distribution describing Q(t). The likelihood $L_{i,t}(\boldsymbol{\theta}|p_{i,t}, \Delta_{i,t})$ determines the learning of the PDF of parameters $\boldsymbol{\theta}$ from projections, in analogy to Eq. (2).

The bias $\Delta_{i,t}$ is modelled as a normal random variable with zero mean and standard deviation $\sigma_{i,t}^{(hidden)}$:

$$\Delta_{i,t} = z \times \sigma_{i,t}^{(hidden)} = z \times \sigma_i^{(internal)} \times \sqrt{\frac{\eta_t^{(hidden)}}{\eta_t^{(internal)}}}, \tag{7}$$

with $z$ being a standard normal random variable. By modeling all $\Delta_{i,t}$ as a function of the same z, it is assumed that the $\Delta_{i,t}$ are fully dependent within one projection i. This treatment is conservative, since it minimizes the amount of learning from

projected discharges. Due to the large impact of the projection on the bias, it is a better depiction of reality than the assumption of independent $\Delta_{i,t}$ within one projection i. From this follows the likelihood for a complete projection time series $\boldsymbol{p}_i$ as

$$L_i(\boldsymbol{\theta}|\boldsymbol{p}_i) = \int_{-\infty}^{\infty} \left[ \prod_{t=1}^{N} f_{Q(t)|\Theta}(p_{i,t} - z \times \sigma_{i,t}^{(hidden)}|\boldsymbol{\theta}) \right] \times \nu(z) dz, \tag{8}$$

where $\nu$ is the standard normal distribution. Internal variability is included in Eq. (8) naturally via $p_{i,t}$, as is parameter uncertainty, which is a function of the length of projections. The estimate of hidden uncertainty, as from Sect. 2.5, is included via $\sigma_{i,t}^{(hidden)}$. While we are focussing on climate uncertainty here, in principle, any kind of additional uncertainty can be

included via the hidden uncertainty parameter $\sigma_{i,t}^{(hidden)}$ in Eq. (8). Model response uncertainty is included in the combination of the likelihoods $L_i(\boldsymbol{\theta}|\boldsymbol{p}_i)$ from different projections i, as described in the following section.



### 3.3 Accounting for dependency among projections

Individual projections are not independent. Hence, one cannot combine $L_i(\boldsymbol{\theta}|\boldsymbol{p}_i)$ into a joint likelihood $L(\boldsymbol{\theta}|\boldsymbol{p})$ via a simple product over projections $\boldsymbol{p}_i$. Dependence among multiple projections is due to common model biases, be it because they e.g. share code from the same institution or because our understanding of climate processes is not perfect (Knutti et al., 2013; Tebaldi and Knutti, 2007). Consequently, confidence in the prediction variance should not increase linearly with the number of projections in an ensemble. Instead, the ensemble should be seen as consisting of an effective number $I$ of quasi-independent projections (adding independent pieces of knowledge) that is smaller than the ensemble size $M$ (Pennell and Reichler, 2011; Sunyer et al., 2013b). We thus partition the ensemble into $J$ sets of $I$ projections, where $J$ is the integer quotient of $\frac{M}{I}$. For each of these sets, the likelihood function can then be formulated as the product of the likelihoods $L_i^{(j)}(\boldsymbol{\theta}|\boldsymbol{p}_i)$ of the set members, since they are assumed to contain independent information:

$$L^{(j)}(\boldsymbol{\theta}|\boldsymbol{p}) = \prod_{i=1}^{I} L_i^{(j)}(\boldsymbol{\theta}|\boldsymbol{p}_i). \tag{9}$$

Climatological rationale is be applied to determine the division of the ensemble into sets: in line with the concept of effective projections, the projections in each set should be as distinct as possible, adding a maximum of additional information.

Based on their genealogy, we partition the available projections (
Table 1) as follows:

- ▪ When using two sets of five effective projections:
    - ○ Set 1: CLM1, CCLM, REMO2, HadGM, RACMO;
    - ○ Set 2: CLM2, REMO1, REMO3, HadRM, BCM.
- ▪ When using three sets of three effective projections (dropping REMO3):
    - ○ Set 1: CLM1, REMO2, HadRM;
    - ○ Set 2: CLM2, REMO1, HadGM;
    - ○ Set 3: CCLM, RACMO, BCM.

The set likelihood $L^{(j)}(\boldsymbol{\theta}|\boldsymbol{p})$ from Eq. (9) is used to compute the joint set posterior of parameters, $f_{\boldsymbol{\Theta}|\mathbf{Q}(t)}^{(j)}(\boldsymbol{\theta}|\boldsymbol{p})$, in analogy to Eq. (2). The set posteriors are then averaged to result in an overall posterior $f_{\boldsymbol{\Theta}|\mathbf{Q}(t)}(\boldsymbol{\theta}|\boldsymbol{p})$ of learning from projections under climate uncertainty. The averaging over posteriors expresses that we place equal trust in distributions learned from the different sets.

### 3.4 Planning under uncertainty

Protection requirements ('criterions') are based on the $T$-year discharge $q^{(T)}$ (viz. Eq. (1)). Since the estimate of $q^{(T)}$ changes as new data becomes available, the capacity of the flood protection system will be re-evaluated in the future, and possibly be adjusted. The probability that adjustment becomes necessary is determined by the level of uncertainty: The higher the





uncertainty in the future extreme discharges, the more likely it is that an adjustment of the protection system will become necessary in the future. To understand why this is, consider Fig. 3: After initial planning, new discharges are observed (lilac dots). If, as pictured here, the observed discharges are higher than expected, the design flood estimate $q^{(T)}$ will increase (we show $q^{(1)}$ to be able to display observations on the same scale). If the uncertainty is large at the time of initial planning – as

is the case here, visualized with the blue, original PDF – then the additional information from the new observation has a larger weight in predicting future extreme discharges. Note that the PDF displayed here shows the distribution of the flood *estimate* $q^{(1)}$, for which we use the weighted mean of the distribution of annual maximum discharges here. Thus, the 99[th] percentile of the shown PDF does not correspond to $q^{(100)}$. The change in $q^{(T)}$ is larger than if the distribution of extreme discharges had been more informative (i.e. more 'certain', less 'spread out'). In practice, the protection will only be adjusted when a significant

change in $q^{(T)}$ has occurred that cannot be compensated by the freeboard and planning margin present (represented by the 'protection level boundary' from which onwards adjustment is needed). To avoid the need for frequent adjustments and increase robustness, the optimization framework thus recommends a higher planning margin when the system is constructed under higher uncertainty initially, as will become apparent in the results.

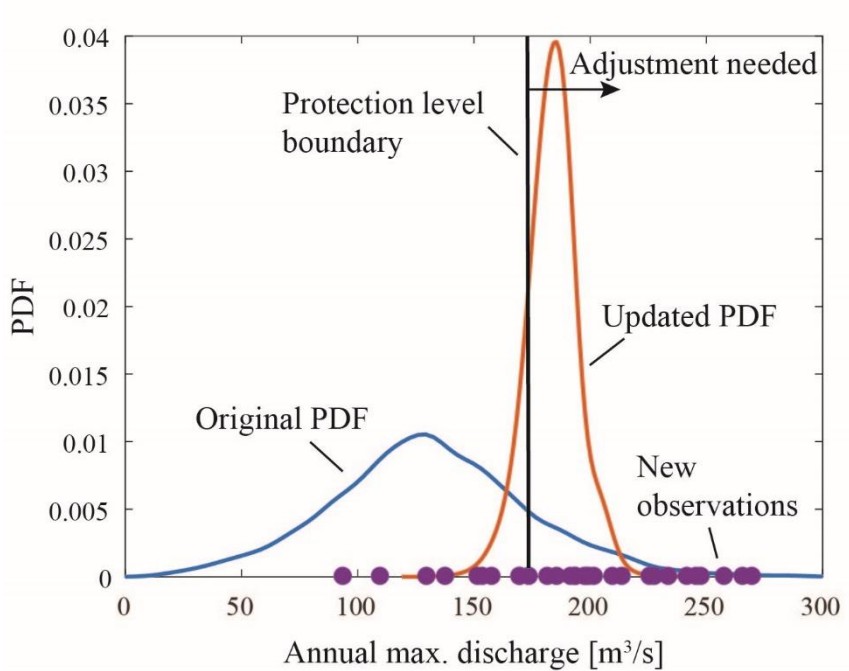

**Figure 3.** Original and updated PDF based on a period of high new observations of annual maximum discharge. Because the original PDF is so broad, the period of extreme observations results in a strongly shifted updated PDF and thus the need for adjusting the protection system.



As has been demonstrated, the – as yet uncertain – future discharge realizations determine future decisions and therefore also have an impact on the optimal initial decision. It is sensible to model protection planning as sequential, with probabilistic future discharge observations, updating of the discharge PDF and corresponding decisions on adjustment in regular time intervals. A Bayesian Network approach doing so for decisions on adapting infrastructure to a changing climate has been presented by (Nishijima, 2015) and a POMDP approach applied to flood protection, using climate scenarios, has been described by (Špačková and Straub, 2016). An alternative sampling-based approach, which takes the full joint parameter PDF into account, has been proposed by (Dittes et al., 2017). The planning horizon is divided into a number of time periods. After each period, the current protection level is re-evaluated and possibly adjusted based on the annual maximum discharges that have been observed during that period – or more precisely, based on the $q^{(100)}$ as resulting from the updated distribution of annual maximum discharges. To probabilistically model this future updating (before these data are actually available), future realizations of annual maximum discharge $\boldsymbol{q}$ are sampled from the discharge distribution $f_{Q(t)|\boldsymbol{\theta}}(q|\boldsymbol{\theta})$ learned initially. Optimal decisions are then identified via backwards induction optimization (Raiffa and Schlaifer, 1961), which works by first determining the system that should be installed at the last adjustment, conditional on the existing protection and discharges observed by then. The obtained recommendation is then used to find the system that should be installed at the second to last adjustment and so forth until arriving at a recommendation for the system that should be installed initially. We employ this optimization framework in the following case study.

## 4 Case study

We present the integration of the uncertainty quantification of extreme discharge in the pre-alpine Mangfall gauge at Rosenheim as shown in Sect. 2 with the uncertainty combination methodology of Sect. 3 and the decision framework of (Dittes et al., 2017). Sect. 4.1 gives details of the implementation, followed by the protection recommendation and sensitivity results in Sect. 4.2.

### 4.1 Implementation

We conduct our case study for the Mangfall river in Rosenheim, which has been introduced in Sect. 2.1. We consider the designed flood protection systems to have a lifetime of 90 years and to be designed such as to protect from the 100-year flood, with design discharge $q^{(100)}$. The decision on the protection capacity will be revised every 30 years, taking into account the discharge records that will be available at these points in time. When learning climate parameters – especially trends – from a time step, 30 years is an often used compromise between the desire to minimize statistical uncertainty and that to capture recent climate developments (IPCC, 2013; Kerkhoff et al., 2015; Laprise, 2014; Pöhler et al., 2012). The protection requirement corresponds to the maximal required protection during the time step in question. As in (Dittes et al., 2017), we used a square root cost function for the construction/extension of the protection system and a discounting rate of 2 %. In (Dittes et al., 2017), we considered a measure of flexibility which describes how costly it is to adapt measures later in their life time. In this





contribution, we give results for the non-flexible case only, which implies that future adjustments to the system are expensive. Introducing some flexibility into the protection system would lead to lower planning margin results than those obtained here.

Following model plausibility testing on the projections (MacKay, 1992), a GEV distribution is chosen to model the annual maximum discharges. It is described by shape parameter $k$, scale parameter $\beta > 0$ and location parameter μ. We assume a

linear trend in the scale and location parameters, which is a common practice in literature (Coles, 2004; Delgado et al., 2010; Hanel and Buishand, 2011; Maraun, 2013). The scale is expressed as $\beta = \beta_0 + \beta_1 \times t$ and the location as $\mu = \mu_0 + \mu_1 \times t$ (Coles, 2004; Hanel and Buishand, 2011)(Coles, 2004; Hanel and Buishand, 2011)(Coles, 2004; Hanel and Buishand, 2011)(Coles, 2004; Hanel and Buishand, 2011)(Coles, 2004; Hanel and Buishand, 2011)(Coles, 2004; Hanel and Buishand, 2011). Thus, $\boldsymbol{\theta} = (k, \beta_0, \mu_0, \beta_1, \mu_1)$.

The joint PDF of parameters of annual maximum discharge learned from the climate projections is used as the basis for future updating with discharge realizations. To obtain this PDF, the climate projections are learned on a prior that is weakly informative in the 100-year design discharge of the first time step (years 1-30) as by Eq. (4). Computationally, the prior is constructed by uniform sampling of parameters over a large space, computing the respective 100-year flood estimate for the first time step for each sampled parameter vector, and performing rejection sampling to obtain 576,000 samples following Eq.

15    (4).

To find the optimal flood protection considering the full sequential decision process, it is necessary to simulate future discharge data, from which new flood estimates will be learned (viz. Sect. 3.4). For this purpose, we used 300 samples of annual maximum discharge in the period 1-30 years and 70 samples of annual maximum discharge in the period 31-60 years. Using fewer discharge samples in later periods is computationally preferable and still comes with a high accuracy, as the absolute

number of samples in the second period overall is $300 \times 70 = 21,000$. This choice of number of samples lead to a relative error of less than 4 % in the protection recommendation.

### 4.2 Protection recommendation and sensitivity

Figure 4 shows the 100-year discharge PDF (weighted mean) from the initial parameter distribution for the first 30 years of planning when learned from the 39-year long historic record versus ten, five, three and one effective projections of 90-year

length. Ten effective projections corresponds multiplying all posteriors and one effective projection corresponds to averaging all posteriors. For five and three effective projections, we split the projections into sets as given in Sect. 3.3.



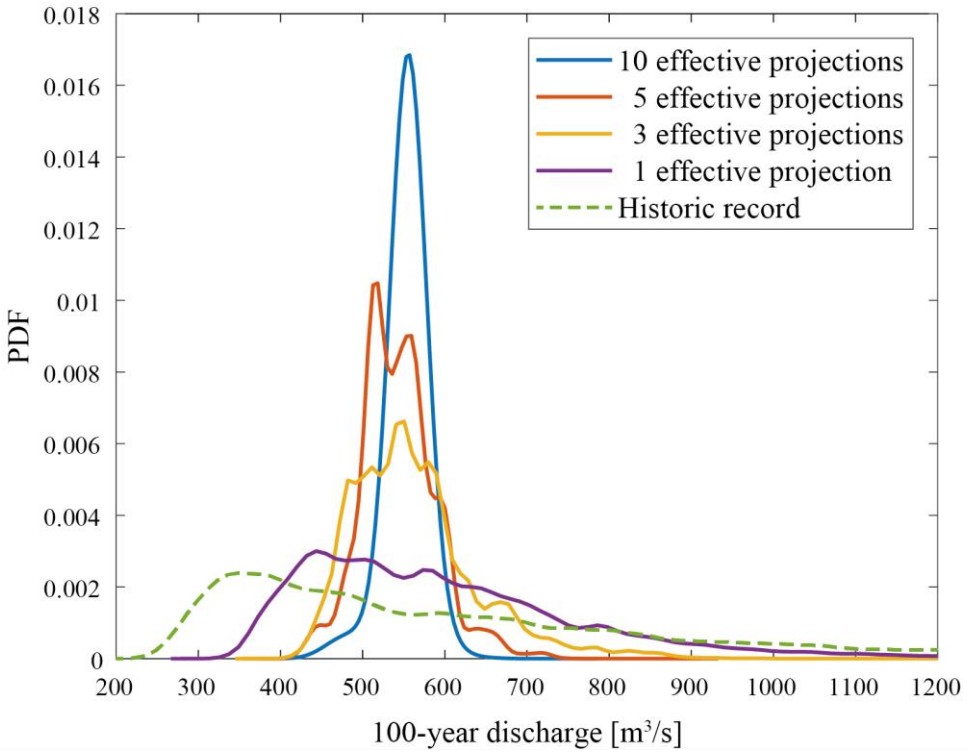

**Figure 4.** 100-year discharge PDF from initial parameter distribution when learned from the historic record (dashed) vs. different numbers of effective projections, for years 1-30.

5   The planning margin that is recommended when learning from the historic record only is 111.8 %, versus 81.9 %, 16.5 %, 12.5 % and 2.6 % for one, three, five, and ten effective projections, respectively. These results are summarized in Table 2.

**Table 2.** Recommended planning margin when using the historic record vs. differing numbers of effective projections for learning the initial parameter space.

| Effective # of projections (or historic) | historic | 1 | 3 | 5 | 10 |
|---|---|---|---|---|---|
| Recommended planning margin [%] | 111.8 | 81.9 | 16.5 | 12.5 | 2.6 |

Using a similar ensemble of climate projections over Denmark, (Sunyer et al., 2013b) established that an ensemble of ten projections corresponds to five effective projections for 20-year heavy summer precipitation. Despite some issues with



transferability – as will be discussed in Chapter 5 – we thus use five effective projections and hence a planning reserve of 12.5 % as the recommended protection margin from the extreme summer precipitation floods observed at the Mangfall in Rosenheim.

To investigate the effect of hidden uncertainty on the protection recommendation, we performed the optimization again, using no hidden uncertainty as well as using double the hidden uncertainty variance shares estimated in Sect. 2.5 (reproduced numerically in Supplement C), with an effective model number of five. The recommended planning margins lay in the expected order, with the 'no uncertainty' recommendation the smallest at 8.1 % and the 'double uncertainty' recommendation the largest at 13.8 %. Finally, we studied the effect of changing the trend in the projections of annual maximum discharge. Detrending the projections lead to a recommendation of 12.2 %. We then used the projections with doubled trend: from the observed average of 0.25 $m^3$/s per year (corresponding to an 11 % rise in mean discharge during the 90-year life-time) to 0.5 $m^3$/s per year. The recommended planning margin increased only very slightly, from 12.5 % to 12.7 %. The results are summarised in Table 3.

**Table 3.** Recommended planning margin [%] when using five effective projections and varying hidden uncertainty and trend.

| Quantity \ Direction of change | none | reference | double |
|---|---|---|---|
| Hidden uncertainty | 8.1 | | 13.8 |
| | | 12.5 | |
| Trend in annual max. discharge | 12.2 | | 12.7 |

## 5 Discussion

It is apparent from the results that the number of effective projections has a large impact on the recommended planning margin. Hence, planners must make use of the concept of effective projections and partition ensembles accordingly, rather than just average over all members of a projection ensembles. Our assumption that five effective projections are applicable for the ten-member ensemble at Rosenheim can be questioned. The transferability of the corresponding results of (Sunyer et al., 2013b) might be hindered by the difference in considered location (a southern German catchment vs. an averaging over Denmark), ensemble (some members differ) and extreme index (100-year event vs 20-year event). From other results presented in (Sunyer et al., 2013b) using an alternative measure of projection dependence as well as higher extreme indices, we believe that the 12.5 % recommendation given here is conservative and a slightly lower recommendation for the planning margin (based on a slightly higher number of effective projections) may be applicable. However, the transferability remains questionable for the location and ensemble and thus the study ideally ought to be repeated for the given catchment and ensemble, in particular with respect to the large impact of the number of effective projections on the protection recommendation.



It is striking that the recommended planning margin from the historic record alone is very large. This is partly since the posterior is sensitive to the assumed extreme value distribution function: we used a GEV distribution with two trend parameters (i.e. five parameters overall) to pick up climate signals in the projections. We are using the same distribution for the historic record for comparability. In reality, one should not attempt to learn such a high number of parameters from such a small set of

data, instead, one would assume stationarity or a fixed trend. We repeated the analysis for a stationary GEV (no trend parameters), resulting in a planning margin recommendation of 75.1 %. This is still high, confirming that it is not recommendable to plan based on a short historic record alone. Additional information should always be used – either, such as done here, projections that have been provided by the climate modelling community and which also incorporate regional information or tools from runoff prediction in ungauged basins, climate analogues, etc. (Arnbjerg-Nielsen et al., 2015; Blöschl

et al., 2013a).

We turn now to the sensitivity analysis. First, the trend: the fact that signals that emerge late on the planning horizon are masked by noise and rendered less relevant by discounting explains why changing the trend signal leads to only insignificant changes in recommended planning margin. This is compounded by the fact that the trend signal is weak, which is to be expected from the location of the case study catchment (Madsen et al., 2014; Maraun, 2013) and is potentially amplified by projections

underestimating trends in extreme precipitation (Haren et al., 2013). It should be added that not all scientists are comfortable with linear trend projections in extreme precipitation and discharge and that there is also an argument to be made for cyclical components (Gregersen et al., 2014) or 'flood-rich' versus 'flood-poor' periods (Hall et al., 2014; Merz et al., 2014), though this may not be applicable to floods of particularly long return periods such as studied here (Merz et al., 2016).

Finally, to the effect of additional uncertainty: we conclude that hidden uncertainty should be considered in decision making

yet when there is already some hidden uncertainty, internal variability and model response uncertainty ('ensemble spread'), further increasing the hidden uncertainty has little effect. This is why we do not engage in detailed discussion on whether the size of the 'hidden uncertainty' has been gauged correctly and whether additional uncertainty components should be included, despite this certainly being debatable (Grundmann, 2010; Refsgaard et al., 2013; Seifert, 2012; Sunyer, 2014; Velázquez et al., 2013). This robustness to additional uncertainty indicates that in the present ensemble, the capacity to project the future

extreme discharge is already extremely limited due to the uncertainty present and thus can barely be reduced by adding more. While this may appear disheartening, it can also be a wake-up call to stop waiting for (doubtful) uncertainty reductions in climate modelling and start taking (robust) decisions (Arnbjerg-Nielsen et al., 2013; Curry and Webster, 2011; Hawkins and Sutton, 2011).



## 6 Conclusions

Estimates of future extreme discharge are fraught with numerous large uncertainties which need to be accounted for in flood protection planning. In particular, the following points must be considered when learning the probability distribution of parameters of extreme discharge (leading to future estimates):

1) an estimate of the uncertainty that can not be quantified from the available data (the 'hidden uncertainty'), since projections and data at hand often cover only a limited range of the uncertainty spectrum (the 'visible uncertainty')

2) the time development of the uncertainty, so as to give less weight to projections far on the projection horizon

3) dependency between projections, since projection ensembles often include several projections sharing code or assumptions

We presented methodology to quantitatively include these aspects when learning the distribution of parameters of annual maximum discharge, e.g. for making flood protection planning decisions. 'Visible' and 'hidden' uncertainty form part of a time-dependent Bayesian likelihood function. Dependence between projections is accounted for by using the concept of effective projection number. We demonstrated the estimation of uncertainties and the application of the optimization framework of (Dittes et al., 2017) in a pre-alpine catchment. The results show that for a given sizable internal variability, the

protection recommendation is robust to further uncertainty and moderate changes in trend. However, uncertainty should not be neglected in planning as this would lead to insufficient protection recommendations. The results also highlight the benefits of combining several sources of information, as has been done here with the different types of uncertainty from climate models and historic records.

**Code availability**

The code is available upon request.

**Data availability**

Data are available in the supplements.

**Author contribution**

B. Dittes developed the concepts of this paper under the guidance of O. Špačková and D. Straub. B. Dittes wrote the code and performed the simulations. The results of Sect. 2.5 were obtained in collaboration of L. Schoppa and B. Dittes. B. Dittes prepared the manuscript with input from all co-authors.





## Competing interests

The authors declare that they have no conflict of interest.

## 5  Acknowledgements

We would like to thank Holger Komischke of Bayerisches Landesamt für Umwelt (LfU) for fruitful discussions. The LfU also provided the discharge records and projections used in the case study. The discharge projections were modelled within the cooperation KLIWA and the Interreg IV B Project AdaptAlp. They were based either on ENSEMBLES data funded by the EU FP6 Integrated Project ENSEMBLES (contract number 505539), whose support is gratefully acknowledged, or additional available climate projections. These are REMO1 ('UBA') and REMO2 ('BfG') (Umweltbundesamt, 2017), as well as CLM1 and CLM2 (Hollweg et al., 2008). This work was supported by Deutsche Forschungsgemeinschaft (DFG) through the TUM International Graduate School of Science and Engineering (IGSSE).

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
