# Peer review of "Managing uncertainty in flood protection planning with climate projections"

_Hydrology and Earth System Sciences, 2017_

## Referee Comment (RC1) · Anonymous Referee #1 · 30 Nov 2017

General comments:

The manuscript presents a Bayesian approach to quantify the uncertainties of the flood protection levels required subject to climate change. The societal benefit of such an approach is obvious, since it helps to improve resilience against increasing flood peaks in a changing climate. The authors focus on the distinction between 'visible' and 'hidden' uncertainties. They provide an approach how to estimate and combine the two sources of uncertainty. The practical relevance is stressed by a case study. The paper demonstrates the strengths of using a Bayesian approach for the quantification of future uncertainties in peak flows. However, it is quite hard to follow the reasoning of the authors. Reordering certain sections and paragraphs may help to improve the quality of the manuscript. Also, the writing should be improved, in particular in sharpening the logical structure.

[Figure]

Specific comments:

- General: It is difficult to follow the different steps and methods applied. Including an additional figure / flow chart illustrating the whole processing chain may help to understand the methods.

- Abstract: What is the goal of the paper? Is it introducing a new framework to model uncertainties in extreme discharges?

- Introduction: The authors thoroughly motivate the need for the study at hand, but they do not introduce the Bayesian decision making framework in enough detail. References and literature on quantitative Bayesian decision making should be added. Further, the choice of the method should be motivated based on a comparison with similar competing approaches.

- Section 2: Subsection 2.1 and 2.2 should be a separate section devoted to the description of the catchment considered and the data only.

- Section 2: Move Subsections 2.3 to 2.6 to Section 3 in order to gather all methods in a single section and hence improving readability.

- Section 2: Subsection 2.4 is too long and does not contain any new scientific findings. Should be shortened.

- Section 2: Subsection 2.5 How are the relative contributions of the different sources of uncertainty specified? Please clarify.

- Section 3: Subsection 3.1 contains a very general discussion comparing visible and hidden uncertainties. This should be move to the introduction Section 1.

Technical corrections:

- p1/l15: Rephrase. Maybe, a formulation like: "Therefore, planning authorities increasingly incorporate discharge projections into the assessment of future protection needs, rater..."
- p2/l12: Please put only the years into brackets, if the reference is part of the sentence. This should be corrected throughout the manuscript.

- p2/l17-23: Difficult to understand. Please clarify the framework in more detail.

- p2/l27-31: Paragraph out of sync. Either explain it in more detail here or move it to Section 3.

- p4/l11: What about rain on snow events?

- p6/l5-6: Why?

- p8/l1-6: How are the parameters estimated in this study? Is this source of uncertainty considered?

- p8/l20&29: Please provide another reference that is publicly available.

- Figure 2: Add a 2nd panel containing the same plot in absolute units in order to avoid misleading conclusions on the changes in uncertainties over time

- p10/l4&11: colons not needed

- p10/l16: viz. is a unusual abbreviation. Do you mean see/refer to? Please check this throughout the manuscript.

- p12/l9: the time index t=1,..,N is confusing. Here it denotes future time steps, in Section 2.6 t=1,...,N denotes historical years.

- p12/eq7: Do you mean $\Delta_{i,t} \sim \mathcal{N}(0,\sigma_{i,t}^{(hidden)})$ ?

- p13/l11: "is applied"

- p16/l5: "which is common practice in the literature"

- p16/l7-9: duplicated citations

- p16/l20&21: Why? The error may be larger, but just not represented by the ensemble of climatological predictions.

- p18/l17: "Hence, we recommend that planners make use. . ."

- p19/l14: Why is this expected?

- p19/l16-18: Could you elaborate a bit more on this topic, since it seems to be related to an important research question.

- p19/l19-21: Difficult to understand, please rephrase.

- p19/l24-25: Please rephrase.

- p20/l10-18: Also a bit difficult to understand, rephrasing it a bit may help.

---

## Referee Comment (RC2) · Anonymous Referee #2 · 11 Dec 2017

General comments

The paper presents a Bayesian methodology to quantify and combine different uncertainty sources for estimation of probability distributions of design discharge under climate change. This is combined with an optimisation framework to derive optimal flood protection. The methodology is demonstrated on a case study.

The presentation of the methodology is difficult to follow, which makes the interpretation of the results problematic. The quantification of the different uncertainty sources, which is a central part of the methodology is only very briefly described (Section 2.5). In addition, for the optimisation framework applied reference is made to an unpublished paper by the same authors, and it is difficult to grasp from the description given in the paper.

Detailed comments

1. The title of the paper is not very informative.

2. Main results should be summarised in the abstract.

3. Section 2.3. In the explanation of internal variability it is stated that "it cannot be predicted with certainty what amount of discharge will be recorded on a given day". But is this an issue here? Internal variability should be related to the problem of estimation of a design discharge. It is also stated that the internal variability is the dominant source of uncertainty, but no documentation for this statement is provided.

4. Section 2.4, p. 8, l. 5-6. It is stated that "the error from the hydrological model is small, in particular for high flow indicators (Velazquez et al., 2013)". Velazquez et al. (2013) conclude that high flow indicators are less sensitive to the choice of hydrological model. This is not to say that the uncertainty in the simulation of extreme discharge events is small. Often you see quite large uncertainties in the simulation of extremes. This can be quantified from the hydrological model simulation in the case study.

5. Section 2.5. This section needs to be elaborated. There is very little explanation of how the different error sources are estimated.

6. Section 4.2. The results are difficult to interpret. The relation between the estimated 100-year pdfs and the planning margins in Table 2 is not clear. It does not seem that the planning margins correspond to the current estimate of the 100-year design discharge of 480 m3/s.

7. Section 4.2, p. 18, l. 9-11. Does this trend relate to the mean discharge? I would expect this trend to be different from the trend of annual maximum discharge.

---

## Author Comment (AC1) · 13 Jan 2018

Please find detailed changes and remarks pertaining to the comments of referee #1 in the supplement. In addition, we would like to make the following general statement:

We have read the comments of the referees with interest. It is apparent that the research is welcomed, yet its presentation needs more clarity. We have used this opportunity to make significant changes to the manuscript based on the reviewer's comments. This was done by incorporating the individual comments – as detailed in the replies to the referees – as well as by doing a 'bird's eye' revision for clarity and coherence.

In the following, we address the two main points/themes raised in both reviews, and which are central to the understanding of the our work.

[Figure]

1) The context and the goals of our research were not sufficiently clear. This is evident from multiple comments made by both reviewers. Re-reading the original manuscript, this has become evident to us. Already the title was not sufficiently clear, and it is understandable that the reviewers partly expected something else than what we present. For this reason, we modify the title of manuscript to "Managing uncertainty in flood protection planning with climate projections", to avoid the impression that the main goal of the paper is the quantification of the uncertainties, which seemed to have been the understanding of the reviewers. Instead, the paper should provide a methodology for dealing in a consistent manner with uncertainties in the context of engineering decision making on flood protection. We clarify these goals in various places of a revised manuscript, starting with an explicit statement in the abstract ("Specifically, we devise methodology to account for uncertainty associated with the use of discharge projections, ultimately leading to planning implications.").

2) There was an unfortunate oversight on our part concerning the utilized Bayesian decision framework, which we proposed previously in (Dittes et al., 2017). That paper (which is still under review), was not available during the discussion, as was criticized by both reviewers. The paper can be downloaded here: era.bgu.tum.de/fileadmin/w00bkd/www/Papers/2017_Dittes_managing_uncertainty.pdf. The reviewers also asked for a more detailed description of the decision framework within the manuscript. We comply with this request by adding an extensive paragraph on the framework in the introduction. Furthermore, we add additional explanation in the case study, and re-phrase various sentences throughout the paper.

We believe that these changes, together with the multiple modifications done in response to the reviewer's detailed comments, improve clarity and will enhance the impact of the paper.

Sincerely,

Beatrice Dittes in the name of all co-authors.

[Figure]

Please also note the supplement to this comment:
https://www.hydrol-earth-syst-sci-discuss.net/hess-2017-576/hess-2017-576-AC1-supplement.pdf
* * *
[Figure]

**Supplement:**

**Author comment on the comment of anonymous referee #1**

The authors thank the referee for the thoughtful and detailed comments. In the following, we respond to the individual suggestions, with referee comments highlighted in *blue*.

> *General: It is difficult to follow the different steps and methods applied. Including an additional figure / flow chart illustrating the whole processing chain may help to understand the methods*

We rephrased p2/l22-26 such that they form a separate, longer paragraph and included a flowchart:

*"In this paper, we show how to incorporate into the flood planning process the visible uncertainty from an ensemble of climate projections as well as hidden uncertainties that can not be quantified from the ensemble itself but may be estimated from literature. When combining uncertainties, special care is taken to account for uncertainty and bias in projections as well as dependencies among different projections. We provide reasoned estimates of climatic uncertainties for a pre-alpine catchment, followed by an application of the previously proposed Bayesian decision framework, sensitivity and robustness analysis. The process is shown in Fig. 1: 1) Projections of annual maximum discharges (see Sect. 2.2) and 2) an estimate of the shares of various uncertainties that are not covered by the projection ensemble (see Sect. 2.5) form the inputs to the analysis. 3) For each projection individually, a likelihood function of annual maximum discharge is computed. This is done such that bias is integrated out and projections later on the horizon are assigned diminuishing weights, making use of the hidden uncertainty shares (see Sect. 3.2). 4) The likelihoods of individual projections are combined using the method of effective projections* (Pennell and Reichler, 2011; Sunyer et al., 2013) *in order to account for dependencies among them (see Sect. 3.3). 5) The Bayesian decision framework of Dittes et al. (2017) is used to obtain 6) a protection recommendation based on the likelihood of extreme discharge. The qualitative basis of the framework is outlined in Sect. 3.4."*

[Figure]

**Figure 1.** Process of finding the recommended planning margin from projections and hidden uncertainty estimate.

> *Abstract: What is the goal of the paper? Is it introducing a new framework to model uncertainties in extreme discharges?*

We have realized that we were not sufficiently clear on our goals in the original manuscript. The main goal of the paper is to introduce a methodology that allows dealing with uncertainties in extreme discharges in practical applications, where data and model availability is limited. As part of this, we present a method to model uncertainties in such situations.

We now made multiple changes to the manuscript to clarify the goals, including a change of the title. This is discussed in our general author comment to all reviewers.

> *Introduction: The authors thoroughly motivate the need for the study at hand, but they do not introduce the Bayesian decision making framework in enough detail. References and literature on quantitative Bayesian decision making should be added. Further, the choice of the method should be motivated based on a comparison with similar competing approaches.*

The corresponding passage has been revised to incorporate the comments of the referee as follows: *"We have previously proposed a fully quantitative Bayesian decision making framework for flood protection (Dittes et al., 2017). Bayesian techniques are a natural way to model discharge probabilistically (Coles et al., 2003; Tebaldi et al., 2004). They also make it easy to combine several sources of information (Viglione et al., 2013). Furthermore, Bayesian methods support updating the discharge distribution in the future, when new information becomes available (Graf et al., 2007). Our framework probabilistically updates the distribution of extreme discharge with hypothetical observations of future discharge, which are modelled probabilistically. This is an instance of a sequential (or 'preposterior') decision analysis (Benjamin and Cornell, 1970; Davis et al., 1972; Kochendorfer, 2015; Raiffa and Schlaifer, 1961). This enables a sequential planning process, where it is taken into consideration that the measure design may be revised in the future. Furthermore, it naturally takes into account the uncertainty in the parameters of extreme discharge. The output of the framework is a cost-optimal capacity recommendation of flood protection measures, given a fixed protection criterion (such as the 100-year flood). To protect for the 100-year flood is common European practice (Central European Flood Risk Assessment and Management in CENTROPE, 2013) and is also the requirement in the case study."*

> *Section 2: Subsection 2.1 and 2.2 should be a separate section devoted to the description of the catchment considered and the data only.*

Yes and no: On the one hand, this would increase structure. On the other hand, the description of catchment and data (Subsections 2.1 and 2.2) is closely linked to the discussion of uncertainties in the catchment and data (Subsections 2.3 to 2.6). Furthermore, a Section consisting only of Subsections 2.1 and 2.2 seems quite short.

> *Section 2: Move Subsections 2.3 to 2.6 to Section 3 in order to gather all methods in a single section and hence improving readability.*

This ties in with the previous point: we intend Section 2 to be a predominantly qualitative description of uncertainties in extreme discharge, with a focus on the catchment at hand. There are no novel methods contained (other than the suggested prior transformation of Eq. (4), which we do not feel merits shifting the entire Subsection). As such, we feel that the logical structure

of the paper is best served by leaving Subsections 2.3 to 2.6 together, preferably, as in the originally proposed paper, together with 2.1 and 2.2 (see previous point).

> *Section 2: Subsection 2.4 is too long and does not contain any new scientific findings. Should be shortened.*

Was shortened.

> *Section 2: Subsection 2.5 How are the relative contributions of the different sources of uncertainty specified? Please clarify.*

These are ball-park figures, based only on the sources and considerations already specified in Subsection 2.5. The results of the case study show that the sensitivity of the planning recommendation to variations in uncertainty is low (see Sections 4.2 and 5), thus an exact quantification is not necessary. We added a sentence to clarify thi*s: "Note that this is done as a rough estimate, since uncertainty quantification is not the focus of this paper. As will become clear in Sect. 4.2 and 5, an exact quantification is also not necessary for the proposed decision making process."*

> *Section 3: Subsection 3.1 contains a very general discussion comparing visible and hidden uncertainties. This should be move to the introduction Section 1.*

Since the uncertainty categorization is the starting point of the novel methodology, we would prefer to leave it as Subsection 3.1.

> *p1/l15: Rephrase. Maybe, a formulation like: "Therefore, planning authorities increasingly incorporate discharge projections into the assessment of future protection needs, rater. . ."*

We changed this sentence as suggested. (We assume the referee meant line 25).

> *p2/l12: Please put only the years into brackets, if the reference is part of the sentence. This should be corrected throughout the manuscript.*

Was changed throughout. (We assume the referee meant line 2).

> *p2/l17-23: Difficult to understand. Please clarify the framework in more detail.*

Yes, see the answer to the comment on the introduction.

> *p2/l27-31: Paragraph out of sync. Either explain it in more detail here or move it to Section 3.*

Was moved to Section 3.4.

> *p4/l11: What about rain on snow events?*

As per analysis of the available discharge record as well as of accounts of large historic floods,

these play a minor role in Rosenheim.

This assumption was made based on the available projections and we have re-phrased the sentence to clarify this: *"In the available projections, the absolute amount of internal variability did not change in time significantly and was thus assumed to be stationary."* The assumption is not necessary: the methodology presented in the paper can just as well be used with non-stationary internal variability.

*p8/l1-6: How are the parameters estimated in this study? Is this source of uncertainty considered?*

They are assumed to be part of the hydrological model uncertainty and thus enter the hidden uncertainty as a ball-park figure (see Section 2.5). The results of the case study show that the sensitivity of the planning recommendation to variations of the hidden uncertainty is low (see Sections 4.2 and 5), thus an exact quantification of its components is not necessary.

*p8/l20&29: Please provide another reference that is publicly available.*

The graphs provided to us by Prof. Hawkins closely resemble those shown in Fig. 11-8 of (IPCC, 2013) and on his website
http://climate.ncas.ac.uk/research/uncertainty/precip/plots.html, where one can also download the papers outlining the methodology (Hawkins and Sutton, 2009, 2011). With the consent of Prof. Hawkins, it may be possible to provide the data in a supplement to this paper.

*Figure 2: Add a 2nd panel containing the same plot in absolute units in order to avoid misleading conclusions on the changes in uncertainties over time*

We produced and included the requested 2$^{nd}$ panel below. Note that the absolute values differ from projection to projection, we used CCLM here. The description was updated as follows: *"The results of the estimation are shown in Fig. 3. Figure 3 (a) shows the resulting relative uncertainty shares and Fig. 3(b) the resulting absolute uncertainties for the projection CCLM"* Note that due to adding the flow chart, Figure 2 became Figure 3.

[Figure]

**Figure 3.** (a) Share of different uncertainty components (variance) for extreme discharge in Rosenheim. (b) Resulting absolute uncertainties for CCLM. Uncertainties that are 'visible' in our case study are shaded yellow/orange, 'hidden' ones blue/green.

*p10/l4&11: colons not needed*

This seems to be a matter of taste, we prefer to keep punctuation in sentences with equations.

*p10/l16: viz. is a unusual abbreviation. Do you mean see/refer to? Please check this throughout the manuscript.*

Yes. We replaced this by the more common "see".

*p12/l9: the time index t=1,..,N is confusing. Here it denotes future time steps, in Section 2.6 t=1,. . .,N denotes historical years.*

$t$ denotes years in both cases. We changed the $N$ to $N'$ in Section 3.2 to indicate that the number of years is not necessarily the same.

*p12/eq7: Do you mean $\Delta_{\{i,t\}} \sim \N(0,\sigma_{\{i,t\}}^{(hidden)\})$ ?*

It is true that one could re-formulate the equation to include the normal distribution explicitly. When the notation *N(mean, standard deviation)* is used, the equation given by the referee is correct, though one more commonly uses the notation *N(mean, variance)*. To avoid such confusion, we would prefer to leave the equation as it is.

*p13/l11: "is applied"*

Was changed.

*p16/l5: "which is common practice in the literature"*

Was changed.

*p16/l7-9: duplicated citations*

Was changed.

*p16/l20&21: Why? The error may be larger, but just not represented by the ensemble of climatological predictions.*

The fact that the ensemble may only reflect part of the climate system is accounted for by the hidden uncertainty. One may think of more error sources, but the results of the case study show that adding further uncertainty does not significantly change the planning recomendation.

*p18/l17: "Hence, we recommend that planners make use. . ."*

Was changed.

*p19/l14: Why is this expected?*

Because the cited papers have found this.

*p19/l16-18: Could you elaborate a bit more on this topic, since it seems to be related to an important research question.*

We appended the following sentence to the paragraph in question: *"We asumed a linear trend in the case study for simplicity, but the proposed methodology is general. To use a different trend representation, one just has to change the definition of θ (see Sect. 4.1) accordingly."*

*p19/l19-21: Difficult to understand, please rephrase.*

We rephrased as follows: *"Finally, we discuss the impact of varying size of uncertainty on planning. To investigate this, we evaluated the recommended planning margin when not adding any hidden uncertainty, when using the estimated amount and when using double the estimated amount of hidden uncertainty (see Sect. 4.2). The effect was small, in particular between adding the estimate vs. double the estimate of hidden uncertainty. The share of hidden uncertainty is larger in the farther future, where its effect is limited because of discounting. We conclude that hidden uncertainty should be considered in decision making, yet the sensitivity to its exact amount is low and when there is already a considerable level of uncertainty, including more has little effect."*

*p19/l24-25: Please rephrase.*

We rephrased as follows: *"We believe that the low sensitivity of the protection recommendation to the size of the hidden uncertainty in the presented case study can be explained by the considerable visible uncertainty present: the capacity to project the future extreme discharge is already extremely limited and can barely be reduced by adding more uncertainty."*

*p20/l10-18: Also a bit difficult to understand, rephrasing it a bit may help*

Rephrased as follows: *"In the proposed methodology, we quantitatively include these aspects in learning the probabilistic distribution of flood discharge. Both 'visible' and 'hidden'*

*uncertainty are included in a time-dependent Bayesian likelihood function. Dependence between projections is accounted for by using the concept of effective projection number. The uncertainty analysis proposed in this paper was used with the optimization framework of (Dittes et al., 2017) to find protection recommendations for a pre-alpine case study catchment. The results show that when there is sizable visible uncertainty, the protection recommendation is robust to further uncertainty and moderate changes in trend. However, hidden uncertainty should not be neglected in planning as this would lead to insufficient protection recommendations."*

**References**

Benjamin, J. R. and Cornell, C. A.: Probability, Statistics and Decisions for Civil Engineers, Mc Graw - Hill Book Company, New York City., 1970.

Central European Flood Risk Assessment and Management in CENTROPE: Current standards for flood protection., 2013.

Coles, S., Pericchi, L. R. and Sisson, S.: A fully probabilistic approach to extreme rainfall modelling, Journal of Hydrology, 273, 35–50, doi:10.1016/S0022-1694(02)00353-0, 2003.

Davis, D. R., Kisiel, C. C. and Duckstein, L.: Bayesian decision theory applied to design in hydrology, Water Resources Research, 8(1), 33–41, 1972.

Dittes, B., Špačková, O. and Straub, D.: Managing uncertainty in design flood magnitude: Flexible protection strategies vs. safety factors, Journal of Flood Risk Management, submitted [online] Available from: https://www.era.bgu.tum.de/fileadmin/w00bkd/www/Papers/2017_Dittes_managing_uncertainty.pdf, 2017.

Graf, M., Nishijima, K. and Faber, M.: Bayesian updating in natural hazard risk assessment, Australian Journal of Structural Engineering, 2007.

Hawkins, E. and Sutton, R.: The potential to narrow uncertainty in regional climate predictions, Bulletin of the American Meteorological Society, 90(8), 1095–1107, doi:10.1175/2009BAMS2607.1, 2009.

Hawkins, E. and Sutton, R.: The potential to narrow uncertainty in projections of regional precipitation change, Climate Dynamics, (37), 407–418, 2011.

IPCC: Climate Change 2013: The Physical Science Basis., 2013.

Kochendorfer, M. J.: Decision Making Under Uncertainty, The MIT Press, Cambridge, Massachusetts., 2015.

Pennell, C. and Reichler, T.: On the effective number of climate models, Journal of Climate, 24(9), 2358–2367, doi:10.1175/2010JCLI3814.1, 2011.

Raiffa, H. and Schlaifer, R.: Applied Statistical Decision Theory, 5th ed., The Colonial Press,

Boston., 1961.

Sunyer, M. A., Madsen, H., Rosbjerg, D. and Arnbjerg-Nielsen, K.: Regional interdependency of precipitation indices across Denmark in two ensembles of high-resolution RCMs, Journal of Climate, 26(20), 7912–7928, doi:10.1175/JCLI-D-12-00707.1, 2013.

Tebaldi, C., Smith, R., Nychka, D. and Mearns, L.: Quantifying uncertainty in projections of regional climate change: a Bayesian approach to the analysis of multimodel ensembles, Journal of Climate, 18, 1524–1540, 2004.

Viglione, A., Merz, R., Salinas, J. L. and Blöschl, G.: Flood frequency hydrology: 3. A Bayesian analysis, Water Resources Research, 49(2), 675–692, doi:10.1029/2011WR010782, 2013.

---

## Author Comment (AC2) · 13 Jan 2018

Please find detailed changes and remarks pertaining to the comments of referee #2 in the supplement. In addition, we would like to make the following general statement (same as for referee #1):

We have read the comments of the referees with interest. It is apparent that the research is welcomed, yet its presentation needs more clarity. We have used this opportunity to make significant changes to the manuscript based on the reviewer's comments. This was done by incorporating the individual comments – as detailed in the replies to the referees – as well as by doing a 'bird's eye' revision for clarity and coherence.

In the following, we address the two main points/themes raised in both reviews, and which are central to the understanding of the our work.

[Figure]

1) The context and the goals of our research were not sufficiently clear. This is evident from multiple comments made by both reviewers. Re-reading the original manuscript, this has become evident to us. Already the title was not sufficiently clear, and it is understandable that the reviewers partly expected something else than what we present. For this reason, we modify the title of manuscript to "Managing uncertainty in flood protection planning with climate projections", to avoid the impression that the main goal of the paper is the quantification of the uncertainties, which seemed to have been the understanding of the reviewers. Instead, the paper should provide a methodology for dealing in a consistent manner with uncertainties in the context of engineering decision making on flood protection. We clarify these goals in various places of a revised manuscript, starting with an explicit statement in the abstract ("Specifically, we devise methodology to account for uncertainty associated with the use of discharge projections, ultimately leading to planning implications.").

2) There was an unfortunate oversight on our part concerning the utilized Bayesian decision framework, which we proposed previously in (Dittes et al., 2017). That paper (which is still under review), was not available during the discussion, as was criticized by both reviewers. The paper can be downloaded here: era.bgu.tum.de/fileadmin/w00bkd/www/Papers/2017_Dittes_managing_uncertainty.pdf. The reviewers also asked for a more detailed description of the decision framework within the manuscript. We comply with this request by adding an extensive paragraph on the framework in the introduction. Furthermore, we add additional explanation in the case study, and re-phrase various sentences throughout the paper.

We believe that these changes, together with the multiple modifications done in response to the reviewer's detailed comments, improve clarity and will enhance the impact of the paper.

Sincerely,

Beatrice Dittes in the name of all co-authors.

Please also note the supplement to this comment:
https://www.hydrol-earth-syst-sci-discuss.net/hess-2017-576/hess-2017-576-AC2-supplement.pdf

————————————————————
[Figure]

**Supplement:**

**Author comment on the comment of anonymous referee #2**

The authors would like to thank the referee for the comments. In the following, we respond to the individual suggestions, with referee comments highlighted in *blue*.

> *The quantification of the different uncertainty sources, which is a central part of the methodology is only very briefly described (Section 2.5). / Section 2.5. This section needs to be elaborated. There is very little explanation of how the different error sources are estimated.*

Uncertainty quantification is not one of the main goals of the paper. As we discuss in our general author comment, we realize that we were not sufficiently clear about the goals of the paper, which causes this misunderstanding. This we have improved in a revised version of the manuscript. In fact, the numbers given in Sect. 2.5 are ball-park figures, based only on the sources and considerations presently stated there. Furthermore, the results of the case study show that the sensitivity of the planning recommendation to variations in uncertainty is low (see Sections 4.2 and 5), thus an exact quantification is not necessary. We added a sentence to clarify this: *"Note that this is done as a rough estimate, since uncertainty quantification is not the focus of this paper. As will become clear in Sect. 4.2 and 5, an exact quantification is also not necessary for the proposed decision making process."* We also enhanced the statement of goals in the abstract (see respective comment).

> *... for the optimisation framework applied reference is made to an unpublished paper by the same authors, and it is difficult to grasp from the description given in the paper.*

We included a download link to the cited paper. Furthermore, we improved the description of the framework and the overall methodology in various places (see revised paper). In particular, the previously brief description in Sect. 1 was extended as follows:

*„We have previously proposed a fully quantitative Bayesian decision making framework for flood protection (Dittes et al., 2017). Bayesian techniques are a natural way to model discharge probabilistically (Coles et al., 2003; Tebaldi et al., 2004). They also make it easy to combine several sources of information (Viglione et al., 2013). Furthermore, Bayesian methods support updating the discharge distribution in the future, when new information becomes available (Graf et al., 2007). Our framework probabilistically updates the distribution of extreme discharge with hypothetical observations of future discharge, which are modelled probabilistically. This is an instance of a sequential (or 'preposterior') decision analysis (Benjamin and Cornell, 1970; Davis et al., 1972; Kochendorfer, 2015; Raiffa and Schlaifer, 1961). This enables a sequential planning process, where it is taken into consideration that the measure design may be revised in the future. Furthermore, it naturally takes into account the uncertainty in the parameters of extreme discharge. The output of the framework is a cost-optimal capacity recommendation of flood protection measures, given a fixed protection criterion (such as the 100-year flood). To protect for the 100-year flood is common European*

*practice (Central European Flood Risk Assessment and Management in CENTROPE, 2013) and is also the requirement in the case study.*

*In this paper, we show how to incorporate into the flood planning process the visible uncertainty from an ensemble of climate projections as well as hidden uncertainties that can not be quantified from the ensemble itself but may be estimated from literature. When combining uncertainties, special care is taken to account for uncertainty and bias in projections as well as dependencies among different projections. We provide reasoned estimates of climatic uncertainties for a pre-alpine catchment, followed by an application of the previously proposed Bayesian decision framework, sensitivity and robustness analysis. The process is shown in Fig. 1: 1) Projections of annual maximum discharges (see Sect. 2.2) and 2) an estimate of the shares of various uncertainties that are not covered by the projection ensemble (see Sect. 2.5) form the inputs to the analysis. 3) For each projection individually, a likelihood function of annual maximum discharge is computed. This is done such that bias is integrated out and projections later on the horizon are assigned diminuishing weights, making use of the hidden uncertainty shares (see Sect. 3.2). 4) The likelihoods of individual projections are combined using the method of effective projections* (Pennell and Reichler, 2011; Sunyer et al., 2013) *in order to account for dependencies among them (see Sect. 3.3). 5) The Bayesian decision framework of Dittes et al. (2017) is used to obtain 6) a protection recommendation based on the likelihood of extreme discharge. The qualitative basis of the framework is outlined in Sect. 3.4.*

[Figure]

**Figure 1.** Process of finding the recommended planning margin from projections and hidden uncertainty estimate.*"*

*The title of the paper is not very informative.*

We agree and hence change the title to *"Managing uncertainty in flood protection planning with climate projections"*. This should highlight that the goals is not to quantify uncertainty but to manage its impact in planning.

*Main results should be summarised in the abstract.*

We added the following passage on results: *„The results show that hidden uncertainty ought to be considered in planning, but the larger the uncertainty already present, the smaller the impact of adding more. The recommended planning is robust to moderate changes in uncertainty as well as in trend. In contrast, planning without consideration of bias and dependencies in and between uncertainty components leads to strongly sub-optimal planning*

*recommendations.*" Note that the main goal of the paper is to present methods, not results. We clarified the goal further in the abstract: *"This paper focuses on climatic uncertainty. Specifically, we devise methodology to account for uncertainty associated with the use of discharge projections, ultimately leading to planning implications."*

> *Section 2.3. In the explanation of internal variability it is stated that "it cannot be predicted with certainty what amount of discharge will be recorded on a given day". But is this an issue here? Internal variability should be related to the problem of estimation of a design discharge.*

It is true that it would be more helpful to refer to annual maxima – from which the design discharge is estimated – rather than days. We rephrased as *„…even with perfect knowledge, it cannot be predicted deterministically what the annual maximum discharge of a year will be, and thus how the design flood estimate will change."*.

> *It is also stated that the internal variability is the dominant source of uncertainty, but no documentation for this statement is provided.*

The citatiation is located at the end of the corresponding sentence: (Maraun 2013).

> *Section 2.4, p. 8, l. 5-6. It is stated that "the error from the hydrological model is small, in particular for high flow indicators (Velazquez et al., 2013)". Velazquez et al. (2013) conclude that high flow indicators are less sensitive to the choice of hydrological model. This is not to say that the uncertainty in the simulation of extreme discharge events is small. Often you see quite large uncertainties in the simulation of extremes. This can be quantified from the hydrological model simulation in the case study.*

When evaluating the hidden uncertainty, the question is „how much would the result differ, if a different model had been chosen", hence the results of (Velazquez et al., 2013) are applicable. (It is presently stated as an example of hidden uncertainty in the introduction: *„For example, if the same hydrological model has been used for all projections, then the hydrological model uncertainty is 'hidden', since one effectively has only a single sample of hydrological model output."*) To make this clear also in the sentence highlighted by the reviewer, we rephrase it as *„the error from the choice of hydrological model…"*.

> *Section 4.2. The results are difficult to interpret. The relation between the estimated 100-year pdfs and the planning margins in Table 2 is not clear.*

The 100-year PDFs are the result of the methodology described in the presented paper. Using these as input to the optimization framework (the description of which was revised, see respective point) leads to the recommendations in Table 2. Sect. 3.4 aims to give an intuitive understanding of the relationship between the 100-year PDFs and the recommendations (simply speaking, more spread in the PDF = more uncertainty = higher planning margin). We added a sentence to clarify: *„The PDFs shown in Fig. 5 are used as input to the optimization framework of (Dittes et al., 2017) to obtain recommendations for the planning margin. Sect. 3.4 gave an intuitive understanding of how these relate to the 100-year PDF."*

> *It does not seem that the planning margins correspond to the current estimate of the 100-year design discharge of 480 m3/s.*

We assume that the reviewer wanted to write „*100-year PDFs*" (referring to the Fig. 5, which was formerly Fig. 4) instead of „*planning margins*". The 100-year design discharge of 480 m$^3$/s is the official figure used by the city of Rosenheim based on a GEV-fit of historic annual maxima, without consideration of uncertainties. The accuracy of this number is not evaluated by us. However, what one can see in Fig. 5 is that it is in agreement with the historic data (with a very large uncertainty margin) while the projections overestimate the 100-year discharge. Therefore, we state in Sect. 3.4 (previously in Sect. 1): „*Note that, since there is often a discrepancy between the level of observed past discharge at a specific gauge and the corresponding regional climate projections, we take the commonly used approach (Fatichi et al., 2013; Pöhler et al., 2012) of computing relative rather than absolute values from the climate projections. Here, this means that we find a planning margin γ based on the projection ensemble and uncertainty estimates from literature, which may then be applied to the absolute protection (100-year flood) as estimated from historic records.*"

> *Section 4.2, p. 18, l. 9-11. Does this trend relate to the mean discharge? I would expect this trend to be different from the trend of annual maximum discharge.*

It refers to the annual maxima. We clarified this by replacing „*projections*" with „*projected annual maxima*" in the mentioned lines (before and after these, it is already made explicit).

**References**

Benjamin, J. R. and Cornell, C. A.: Probability, Statistics and Decisions for Civil Engineers, Mc Graw - Hill Book Company, New York City., 1970.

Central European Flood Risk Assessment and Management in CENTROPE: Current standards for flood protection., 2013.

Coles, S., Pericchi, L. R. and Sisson, S.: A fully probabilistic approach to extreme rainfall modelling, Journal of Hydrology, 273, 35–50, doi:10.1016/S0022-1694(02)00353-0, 2003.

Davis, D. R., Kisiel, C. C. and Duckstein, L.: Bayesian decision theory applied to design in hydrology, Water Resources Research, 8(1), 33–41, 1972.

Dittes, B., Špačková, O. and Straub, D.: Managing uncertainty in design flood magnitude: Flexible protection strategies vs. safety factors, Journal of Flood Risk Management, submitted [online] Available from: https://www.era.bgu.tum.de/fileadmin/w00bkd/www/Papers/2017_Dittes_managing_uncertainty.pdf, 2017.

Fatichi, S., Rimkus, S., Burlando, P., Bordoy, R. and Molnar, P.: Elevational dependence of climate change impacts on water resources in an Alpine catchment, Hydrology and Earth System Sciences Discussions, 10(3), 3743–3794, doi:10.5194/hessd-10-3743-2013, 2013.

Graf, M., Nishijima, K. and Faber, M.: Bayesian updating in natural hazard risk assessment, Australian Journal of Structural Engineering, 2007.

Kochendorfer, M. J.: Decision Making Under Uncertainty, The MIT Press, Cambridge, Massachusetts., 2015.

Maraun, D.: When will trends in European mean and heavy daily precipitation emerge?,

Environmental Research Letters, 8(1), 14004, doi:10.1088/1748-9326/8/1/014004, 2013.

Pennell, C. and Reichler, T.: On the effective number of climate models, Journal of Climate, 24(9), 2358–2367, doi:10.1175/2010JCLI3814.1, 2011.

Pöhler, H., Schultze, B. and Scherzer, J.: KLIWA : Vergleichende Analyse der neuen globalen Klimaprojektionen aus CMIP5 für Süddeutschland. Abschlussbericht., 2012.

Raiffa, H. and Schlaifer, R.: Applied Statistical Decision Theory, 5th ed., The Colonial Press, Boston., 1961.

Sunyer, M. A., Madsen, H., Rosbjerg, D. and Arnbjerg-Nielsen, K.: Regional interdependency of precipitation indices across Denmark in two ensembles of high-resolution RCMs, Journal of Climate, 26(20), 7912–7928, doi:10.1175/JCLI-D-12-00707.1, 2013.

Tebaldi, C., Smith, R., Nychka, D. and Mearns, L.: Quantifying uncertainty in projections of regional climate change: a Bayesian approach to the analysis of multimodel ensembles, Journal of Climate, 18, 1524–1540, 2004.

Viglione, A., Merz, R., Salinas, J. L. and Blöschl, G.: Flood frequency hydrology: 3. A Bayesian analysis, Water Resources Research, 49(2), 675–692, doi:10.1029/2011WR010782, 2013.

---

## Author Response (AR2)

**Reply to editor**

The authors would like to thank the editor for his careful reading and technical comments. The technical corrections have all been incorporated, as discussed below, and the revised manuscript has been uploaded.

Yours sincerely,

Beatrice Dittes

on behalf of all co-authors

**Technical corrections**

Editor comments are marked in *blue*.

> *p3/l1-5: The sentence "When combining…" seems to be a bit out-of-sync to me, because you are moving from a paper specific statement in the sentence before back to a very general statement. I would recommend to swap and slightly rephrase the first two sentences on p3.*

The sentence in question is also meant as paper specific, which we clarify by formulating as follows:
*"In this paper, we show how to incorporate into the flood planning process the visible uncertainty from an ensemble of climate projections as well as hidden uncertainties that cannot be quantified from the ensemble itself but may be estimated from literature. In the process of combining these uncertainties, we account for uncertainty and bias in projections as well as for dependencies among different projections."*

> *p6/l4: SRES should be defined here (instead of p7/l25)*

Done.

> *Table 1: Are the indentations (e.g. for "Hydrological model) really needed? I would remove them.*

Done. This was an error in formatting on our part.

> *Table 1: What is R1 to R3?*

We have added the following explanation in the header: *"R1-R3 denote distinct model runs"*.

> *p11/l25: I would move the explanations of aleatory and epistemic uncertainty to Section 2.3, where the word aleatory is used the first time*

We agree that the term should be introduced at its first mention. However, since we think a short discussion of aleatory vs. epistemic uncertainty fits better at p11/l25 than in Section 2.3, we chose to remove the term from the latter, rather than elaborate on it there: "*The term 'internal variability' describes the irreducible uncertainty component in extreme discharge: even with perfect knowledge, it cannot be predicted deterministically […]*"

We feel that the Figure itself is quite full already, but we clarify now in the description and caption that the estimate corresponds to the peak of the PDF:

*"Since the estimate of $q^{(T)}$ – the peak of the PDF – changes as new data becomes available, the capacity of the flood protection system will be re-evaluated in the future, and possibly be adjusted."*

*"**Figure 4.** Original and updated PDF based on a period of high new observations of annual maximum discharge. Because the original PDF is so broad, the period of extreme observations results in a strongly shifted updated PDF, the peak of which (corresponding to the estimate of $q^{(1)}$) crosses the protection boundary level. Thus, the protection system must be adjusted."*

Done.

*Throughout the paper the term "learning" is used in relation to estimation of distribution parameters. "Estimation" is the commonly used term in statistics.*

We modify the text to use „estimation" when referring to the classical statistical estimation of the distribution or model parameters. We still keep the broader term "learning" in some contexts.

*It is not common to include multiplication signs "x" in equations.*

Changed throughout.

*Text above table 3. I expect you refer to trend in mean annual maximum discharge and not mean discharge.*

Yes, was corrected.